# Hyperspectral Modeling of Soil Organic Matter Based on Characteristic Wavelength in East China

**Mingsong Zhao [1,2,3,*], Yingfeng Gao [1,2,3], Yuanyuan Lu [4] and Shihang Wang [1,2,3]**

1 School of Geomatics, Anhui University of Science and Technology, Huainan 232001, China;
gao18321684385@163.com (Y.G.); wangshihang122@sina.com (S.W.)
2 Key Laboratory of Aviation-Aerospace-Ground Cooperative Monitoring and Early Warning of Coal Mining-induced Disasters of Anhui Higher Education Institutes, Huainan 232001, China
3 Coal Industry Engineering Research Center of Collaborative Monitoring of Mining Area's Environment and Disasters, Huainan 232001, China
4 Nanjing Institute of Environmental Sciences, Ministry of Ecology and Environment of the People's Republic China, Nanjing 210042, China; yylu@nies.org
* Correspondence: zhaomingsonggis@163.com

**Abstract:** Soil organic matter (SOM) is a key index of soil fertility. Visible and near-infrared (VNIR, 350–2500 nm) reflectance spectroscopy is an effective method for modeling SOM content. Characteristic wavelength screening and spectral transformation may improve the performance of SOM prediction. This study aimed to explore the optimal combination of characteristic wavelength selection and spectral transformation for hyperspectral modeling of SOM. A total of 219 topsoil (0–20 cm) samples were collected from two soil types in the East China. VNIR reflectance spectra were measured in the laboratory. Firstly, after spectral transformation (inverse-log reflectance (LR), continuum removal (CR) and first-order derivative reflectance (FDR)) of VNIR spectra, characteristic wavelengths were selected by competitive adaptive reweighted sampling (CARS) and uninformative variables elimination (UVE) algorithms. Secondly, the SOM prediction models were constructed based on the partial least squares regression (PLSR), random forest (RF) and support vector regression (SVR) methods using the full spectra and selected wavelengths, respectively. Finally, optimal SOM prediction models were selected for two soil types. The results were as follows: (1) The CARS algorithm screened 40–125 characteristic wavelengths from the full spectra. The UVE algorithm screened 105–884 characteristic wavelengths. (2) For two soil types and full spectra, CARS and UVE improved the SOM modeling precision based on the PLSR and SVR methods. The coefficient of determination ($R^2$) value in the validation of the CARS-PLSR (PLSR model combined with CARS) and CARS-SVR (SVR model combined CARS) models ranged from 0.69 to 0.95, and the relative percent deviation (RPD) value ranged from 1.74 to 4.31. Lin's concordance correlation coefficient (LCCC) values ranged from 0.83 to 0.97. The UVE-PLSR and UVE-SVR models showed moderate precision. (3) The PLSR and SVR modeling accuracies of Paddy soil were better than those for Shajiang black soil. RF models performed worse for both soil types, with the $R^2$ values of validation ranging from 0.22 to 0.68 and RPD values ranging from 1.01 to 1.60. (4) For Paddy soil, the optimal SOM prediction models (highest $R^2$ and RPD, lowest root mean square error (RMSE)) were CR-CARS-PLSR ($R^2$ and RMSE: 0.97 and 1.21 g/kg in calibration sets, 0.95 and 1.72 g/kg in validation sets, RPD: 4.31) and CR-CARS-SVR ($R^2$ and RMSE: 0.98 and 1.04 g/kg in calibration sets, 0.91 and 2.24 g/kg in validation sets, RPD: 3.37). For Shajiang black soil, the optimal SOM prediction models were LR-CARS-PLSR ($R^2$ and RMSE: 0.95 and 0.93 g/kg in calibration sets, 0.86 and 1.44 g/kg in validation sets, RPD: 2.62) and FDR-CARS-SVR ($R^2$ and RMSE: 0.99 and 0.45 g/kg in calibration sets, 0.83 and 1.58 g/kg in validation sets, RPD: 2.38). The results suggested that the CARS algorithm combined CR and FDR can significantly improve the modeling accuracy of SOM content.

**Keywords:** competitive adaptive reweighted sampling algorithm (CARS); uninformative variables elimination (UVE); soil hyperspectral data; soil organic matter; support vector regression

## 1. Introduction

Hyperspectral technology can quickly and easily acquire continuous spectral curves of soils containing various wavelengths and rich spectral information. Such technology can reflect multiple soil properties comprehensively, thus enabling high-efficiency and accurate modeling predictions of soil properties [1–7]. Many studies concerning the applications of visible and near-infrared (VNIR, 400–2500 nm) and mid-infrared reflectance spectroscopy (MIRS, 2500–25,000 nm) technology in modeling predictions of soil properties have been undertaken [8–12].

The traditional soil organic matter (SOM) content test method is complicated and expensive, whereas hyperspectral technology can quickly and accurately test the SOM content [13–16]. Many studies concerning the application of VNIR hyperspectral technology in modeling predictions of soil properties have been undertaken. Various models were used for SOM spectral predictions, such as multiple linear regression [17], partial least squares regression (PLSR) [15,18,19], multivariate adaptive regression spline (MARS) [10,20], artificial neural networks [21], machine learning [10,22,23], deep learning [24] and other methods. Systematic comparisons of modeling methods have also been conducted. Among these models, PLSR has relatively high overall precision and is widely used [10,18,23,25].

Due to differences in the soil forming environment, there are different hyperspectral characteristics among different geographical regions and soil types. Many studies have been conducted in different geographical regions and soil types [10,19–21,26]. SOM prediction models with spectral variables for grouping soil samples are more accurate than global modeling methods. A local PLSR model based on the spatial constraints proposed by Shi et al. [19] predicted the SOM content more accurately using a soil spectral library in China. Bao et al. [26] improved the SOM prediction accuracy by applying an optimal soil grouping strategy.

Soil hyperspectral data are composed of various wavelengths with different correlation degrees among them and some information redundancy. Characteristic wavelength screening aims to eliminate the uninformative variables while selecting characteristic variables from hyperspectral data using algorithms and various criteria. After characteristic wavelength screening, the number of spectral wavelengths is compressed significantly, which reduces the dimensionality of the variables and the complexity of the models in the modeling process. Sophisticated methods include the competitive adaptive reweighted sampling (CARS) algorithm, the uninformative variables elimination (UVE) algorithm, the successive projections algorithm (SPA), uniform-interval wavelength reduction, the genetic algorithm, and particle swarm optimization [26–31]. Moreover, combinations of multiple algorithms, such as UVE-SPA, CARS-SPA and Monte Carlo-based UVE, have been used to optimize selected wavelengths [29,32]. Some studies reported that CARS could compress the number of original spectral wavelengths to lower than 16% [26,32,33].

PLSR modeling precision based on selected characteristic wavelengths—usually higher than that based on the full spectra [26–28,31] or the dimensionality of the spectral data—can be significantly reduced while assuring modeling precision [34]. The CARS and UVE algorithms optimize wavelength selection based on the PLSR model. The UVE algorithm selects characteristic wavelengths based on stability analyses of the PLSR regression coefficient [30]. The CARS algorithm selects characteristic wavelengths with high absolute regression coefficient values in the PLSR model [29]. Both algorithms were shown to be effective ways to reduce the number of inputs and improve the PLSR modeling accuracy of SOM [26,27,32,33]. However, it is rarely reported whether CARS and UVE algorithms can improve the accuracy of machine learning methods, such as random forest (RF), support vector regression (SVR) and others.

Spectral transformation, such as inverse-log reflectance (LR), continuum removal (CR), first-order derivative reflectance (FDR) and fractional order derivative, might increase the precision of SOM prediction models by enhancing the absorption or reflection characteristics of the soils in some wavelengths [11,23,35–37]. For example, Nawar et al. [35] and Dotto et al. [23] reported that CR and FDR transformation had a strong positive influence

on the performance of most SOM prediction models. FDR transformation showed better model performance than the second derivative transformation for SOM estimations in several modeling methods [36]. Some research has also explored the prediction effect of SOM content using characteristic wavelength screening combined with different spectral transformation techniques [38,39]. As shown above, characteristic wavelength screening, spectral transformation and combinations of two means have been widely applied to improve the accuracy of SOM spectral modeling.

Paddy soil and Shajiang black soil, i.e., the two main types of cultivated soil in East China, were selected as the study object in this research. After different spectral transformations of the VNIR hyperspectral data of the two soil types, characteristic wavelength datasets were selected using the CARS and UVE algorithms. Then, PLSR, SVR and RF were used to establish SOM prediction models. The objectives of this research were to: (1) analyze the influence of the CARS and UVE algorithms on the accuracy of the PLSR, SVR and RF models, (2) compare improvements of modeling accuracy by characteristic wavelength screening and spectral transformation, and (3) assess the modeling performance of the PLSR, SVR and RF models and establish an optimal SOM prediction model for Paddy soil and Shajiang black soils in East China.

## 2. Materials and Methods

### 2.1. Study Area

Study area 1 is located in the central plains of Jiangsu Province (119°53′37″–120°14′4″ E, 32°20′17″–32°44′50″ N) in eastern China, covering an area of 1050 km$^2$ (Figure 1). The annual average temperature, precipitation and elevation are 14.5 °C, 991.7 mm and 5–10 m, respectively. Parent materials mainly included lagoonal facies sediments. Paddy soil dominates. Paddy fields dominate land use type, and the rice-rape rotation is the main crop rotation system.

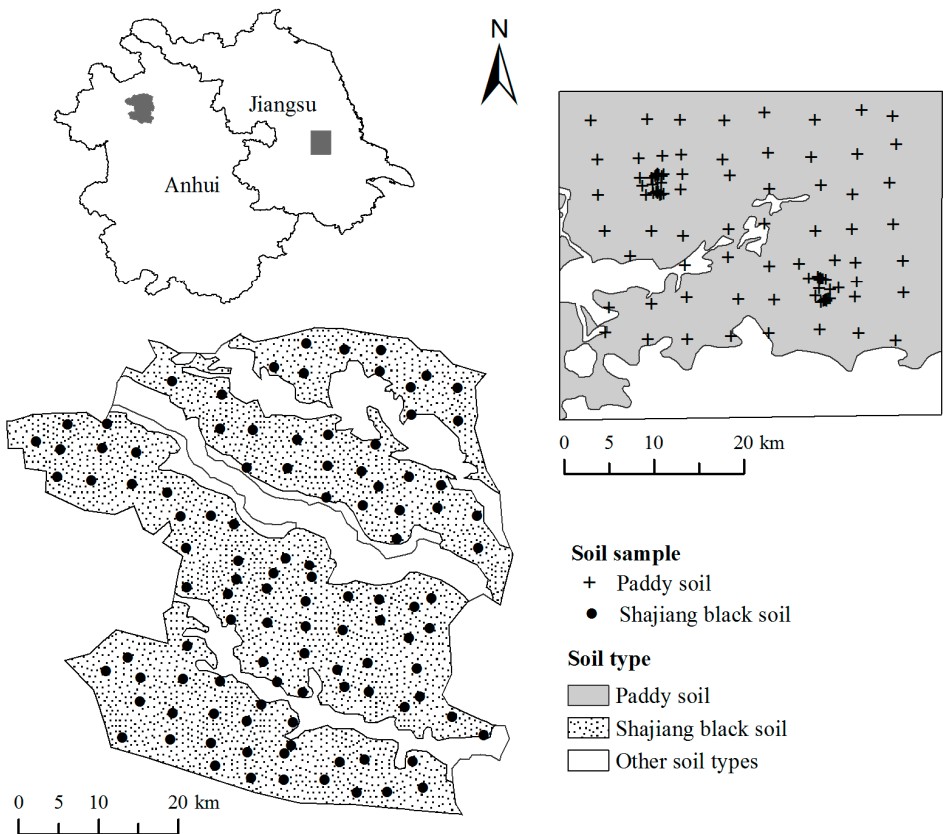

**Figure 1.** Location and distribution of sampling points.

Study area 2 is located in Huaibei Plain of Anhui Province (116°15′43″–116°49′25″ E, 32°55′29″–33°29′64″ N) in eastern China, covering an area of 2091 km$^2$ (Figure 1). The annual average temperature and precipitation are 14.8 °C and 821.5 mm, respectively. The elevation is 20–30 m, decreasing from the northwest to the southeast. Shajiang black soil dominates. Upland dominates the land use type, and wheat-soybean rotation is the main crop rotation system.

### 2.2. Soil Sampling and Analysis

In study area 1, 111 Paddy soil samples were collected from the surface layer (0–20 cm) in November 2009 (Figure 1). In study area 2, 108 Shajiang black soil samples were collected from the surface layer in June 2016 (Figure 1). In each field, 8–12 soil samples were collected within a radius of 10–20 m from the field center. The collected soil samples were mixed and 1 kg was retained using the quartation method. After soil samples were air-dried and ground in the lab, a part of each sample was sieved using a 0.2-mm soil sieve and used to measure SOM content. The SOM content was determined using the potassium dichromate method, which is the same as the wet oxidation method [40].

### 2.3. Soil Spectrum Collection and Preprocessing

After air-drying, grinding and sieving (<2 mm), the diffuse reflectance spectra of the soil samples were measured using an ASD FieldSpec 4 portable spectral radiometer (Analytical Spectral Devices Inc., Boulder, CO, USA). The wavelength range and resampling interval were VNIR (350–2500 nm) and 1 nm, respectively. The entire operation was performed in a dark laboratory with controlled lighting conditions; the light source was a halogen lamp. The soil samples were placed in containers with a diameter of 10 cm and a depth of 1.5 cm, and the surface of the soil sample was scraped flat. The sensor probe was located 15 cm above the surface of the soil sample, with a probe view angle of 25°. A white panel with 99% reflectance was used to calibrate the spectrometer before measuring. Each sample was rotated four times, and 10 scans were performed from each direction. Hence, 40 scanning spectral curves were collected for each sample and the mean was used as the spectra of the soil sample [41].

The Savitzky-Golay (SG) filter method with a moving window of 11 nm and a local polynomial order of 2 regression was used to smooth the reflectance curves. LR, CR and FDR were applied to transform the original reflectance (R) to strengthen the relationship between the SOM and the spectra. Finally, each soil sample yielded 2141 wavelengths for each type of spectra data in the VNIR (355–2495 nm) domain. Spectral data processing was performed using "*prospectr*" package [42] in the R software.

### 2.4. Characteristic Wavelength Screening Algorithms

The CARS algorithm selects characteristics by choosing variables with high absolute regression coefficient values in the PLSR model. It consists of three major steps, i.e., Monte Carlo sampling, PLSR modeling and the acquisition of variable weights. This algorithm executes forced wavelength selection by the exponential damping function and makes competitive wavelength selections using the adaptive reweighted sampling technique. The detailed process of the CARS algorithm is shown in the references [29]. The UVE algorithm is a variable selection approach based on stability analysis of the PLSR regression coefficient [30]. This algorithm eliminates uninformative variables that have relatively small covariance with dependent variables but high variances. The detailed process of the UVE algorithm is shown in the references [43,44]. The CARS and UVE algorithms were applied in MATLAB R2012a.

### 2.5. SOM Spectral Modeling

A total of 111 Paddy soil samples were divided into a calibration set (74, 2/3) and a validation set (37, 1/3) using the Kennard-Stone method. For Shajiang black soil, 72 soil samples were selected as the calibration set, and 36 were used as the validation set. Figure 2

presents a flowchart of the process. Firstly, characteristic wavelengths were screened from R, LR, CR and FDR spectral data using the CARS and UVE algorithms, respectively. Secondly, for each type of spectral data, the SOM models were established based on the PLSR, SVR and RF models using characteristic wavelengths and full spectra. Finally, the performance of the models was compared and the optimal model for each soil type was selected.

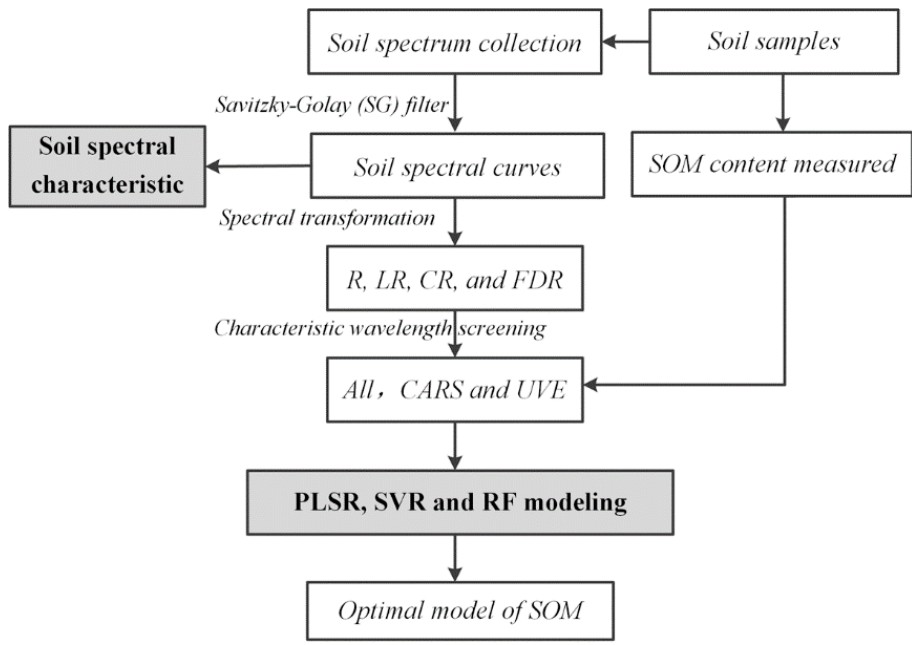

**Figure 2.** Flowchart of the process applied in this research.

In RF modeling, the two main parameters are the number of trees growing in the forest ($n_{tree}$) and the number of randomly selected predictor variables at each node ($m_{try}$). In SVR modeling, the linear kernel function was used to build the model and the main parameter was the penalty coefficient (*C*). The parameters $m_{try}$ and $n_{tree}$ were set to 1–5 and 100–2000 for RF modeling, respectively, and the *C* range was set to $2^{-4}$–$2^4$ for SVR modeling. The "*e1071*" package [45] in the R software was used for parameter tuning using grid search and 10-fold cross-validation. RF and SVR modeling and parameter tuning were performed using the "*e1071*" packages in the R software. PLSR modeling was performed using the "*pls*" package [46] in the R software. Statistical analyses were performed using the "stats" package [47] in the R software (R Core Team, R version 4.2.0, https://www.r-project.org/).

*2.6. Model Evaluation*

The coefficient of determination ($R^2$), root mean square error (*RMSE*), relative percent deviation (*RPD*) and Lin's concordance correlation coefficient (*LCCC*) were chosen as the evaluation indexes. *RMSE* is smaller as $R^2$ approaches 1, indicating better stability and higher prediction precision of the model. If *RPD* is below 1.5, the models have poor estimation abilities. With *RPD* in the range of 1.5 to 1.8, the estimation precision of the models is improved to some extent, but it has a margin for improvement. For *RPD* in the range of 1.8 to 2, the prediction is considered to be good. When *RPD* is higher than 2, the models achieve a high level of precision. *LCCC* represents the distribution and aggregation degree of the predicted and observed values near the 1:1 line; the larger the value, the better.

The calculation formulas of the different evaluation indexes were as follows:

$$R^2 = 1 - \sum_{i=1}^{n}(O_i - P_i)^2 / \sum_{i=1}^{n}(O_i - \overline{O})^2 \tag{1}$$

$$RMSE = \sqrt{\frac{1}{n}\sum_{i=1}^{n}(O_i - P_i)^2} \tag{2}$$

$$RPD = s_o / RMSE \tag{3}$$

$$LCCC = 2rs_o s_p / \left[ s_o{}^2 + s_p{}^2 + \left(\overline{O} - \overline{P}\right)^2 \right] \tag{4}$$

where $O_i$ and $P_i$ are the observed and predicted values, respectively; $\overline{O}$ and $\overline{P}$ are the mean values of observed and predicted values, respectively; $s_o$ and $s_p$ are the corresponding standard deviations; $r$ is the correlation coefficient between the observed and predicted values; and $n$ is the number of observations.

## 3. Results

### 3.1. Characteristic of Soil Spectral Curves

The SOM of Paddy soil samples was relatively high, averaging $32.13 \pm 7.21$ g/kg (Table 1), while that of Shajiang black soil was relatively low, averaging $21.60 \pm 3.94$ g/kg. The coefficients of variation (CV) of SOM in Paddy soil and Shajiang black soil were 18.24% and 22.44%, showing moderate variation. The CV of Paddy soil was relatively high.

**Table 1.** Statistical characteristics of soil organic matter.

| Soil Type | Data Sets | $n$ | Range (g/kg) | Mean (g/kg) | SD [a] | Skewness | Kurtosis | CV [b] (%) |
|---|---|---|---|---|---|---|---|---|
| Paddy soil | All samples | 111 | 15.43~58.22 | 32.13 | 7.21 | 0.50 | 0.92 | 22.44 |
| | Calibration sets | 74 | 15.43~52.49 | 31.93 | 7.03 | 0.23 | 0.16 | 22.01 |
| | Validation sets | 37 | 18.43~58.22 | 32.52 | 7.64 | 0.95 | 2.20 | 23.50 |
| Shajiang black soil | All samples | 108 | 6.65~31.30 | 21.60 | 3.94 | −0.14 | 1.10 | 18.24 |
| | Calibration sets | 72 | 6.65~30.25 | 21.47 | 4.01 | −0.39 | 1.57 | 18.69 |
| | Validation sets | 36 | 15.62~31.30 | 21.84 | 3.82 | 0.46 | −0.08 | 17.50 |

[a] SD, Standard deviation; [b] CV, Coefficient of variation.

The SOM content was divided into seven levels: <15 g/kg, 15–20 g/kg, 20–25 g/kg, 25–30 g/kg, 30–35 g/kg, 35–40 g/kg and >40 g/kg [39]. The mean spectral reflectance curves corresponding to seven SOM content levels were calculated (Figure 3a). With increasing SOM content, the spectral reflectance of the soil decreased over the full spectral range except for the spectral curve of SOM below 15 g/kg. With increasing wavelength, the reflectance in the visible spectrum increased quickly. In the NIR wavelength, the reflectance of soils showed stable growth (Figure 3a). The mean spectral reflectance of soil samples with SOM < 15 g/kg was smaller than that of the samples with SOM from 15 to 25 g/kg.

The absorption characteristics were not apparent in the original spectral curves; however, after CR transformation, they were visibly strengthened and the depth of the absorption valley increased (Figure 3b). Except for the more prominent absorption valleys near 1400 nm, 1900 nm and 2200 nm, the relevant evident characteristics were also detected near 500 nm, 650 nm and 850 nm, respectively. The absorption characteristics near 650 nm were generally strengthened with an increase in SOM content.

The spectral reflectance curves of the two soil types at the minimum, 25%, 50%, 75% and maximum of SOM content were used to analyze the spectral characteristics (Figure 4). The spectral reflectance curves of Paddy soil and Shajiang black soil gradually became flat with increasing SOM content, indicating negative correlation between the spectral reflectance and SOM content (Figure 4). For Shajiang black soil, the two spectral curves when the SOM content was 18.91 g/kg and 21.82 g/kg had extremely similar and overlapping characteristics (Figure 4b). Similar phenomena were observed between the two spectral curves when the SOM content was 23.89 g/kg and 31.30 g/kg (Figure 4b). The spectral features showed no significant differences with the change in SOM content.

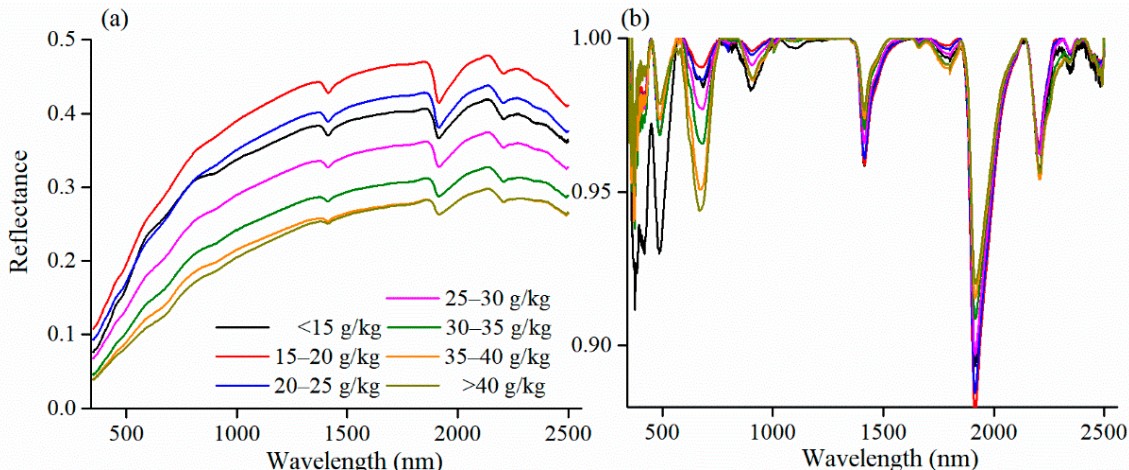

**Figure 3.** Spectral curves of mean soil reflectance (**a**) and continuum removed (**b**) relative to SOM content.

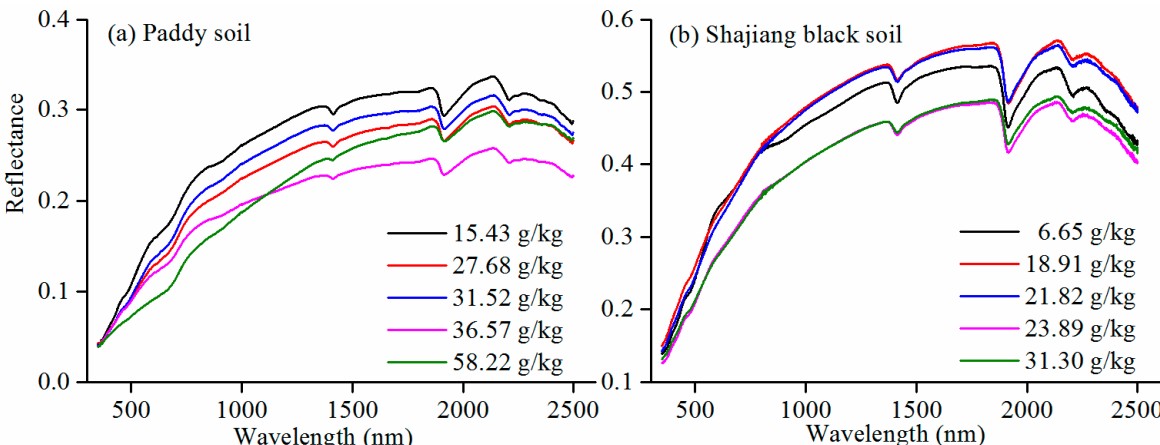

**Figure 4.** Spectral curves of two soil types.

SOM content showed significantly negative correlations with the R spectra in the full spectra range; however, it showed significantly positive correlations with the LR spectra (Figure 5). For Paddy soil, the correlations between SOM content and spectra were stronger. The correlations in the 400–900 nm wavelengths were significantly stronger than those in the other wavelengths and the absolute values of the correlation coefficients were higher than 0.6 (Figure 5a). For Shajiang black soil, slightly weaker correlations with the SOM content were observed, without great differences in correlation among the different wavelengths. The absolute value of the correlation coefficient was between 0.30 and 0.48. The SOM, CR and FDR spectra presented significant positive or negative correlations at 400–750 nm, 1400–1700 nm and 2200–2400 nm wavelengths, and the absolute values of the correlation coefficients were lower than those of the R and LR spectra (Figure 5).

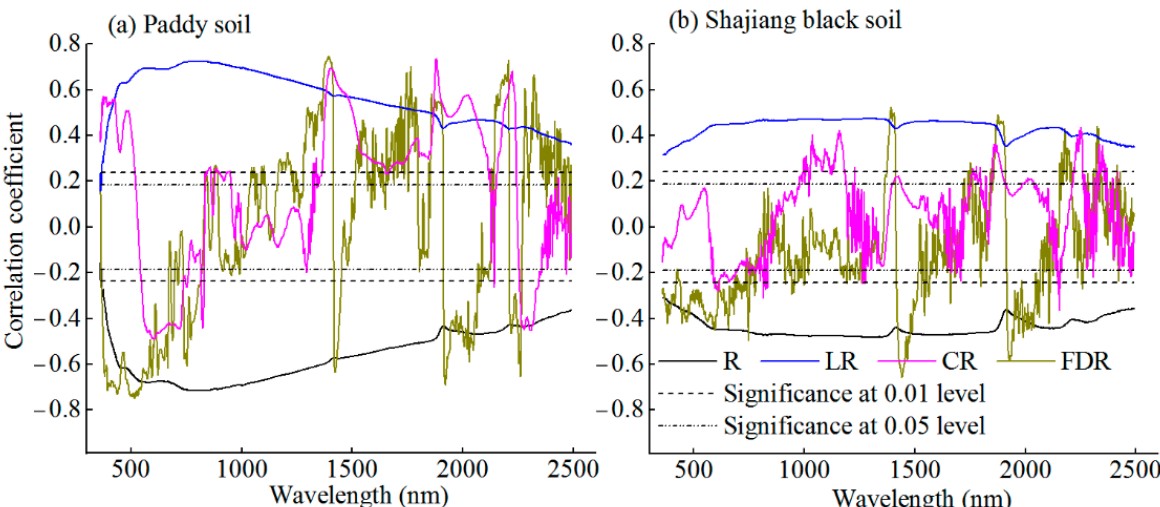

**Figure 5.** Correlation coefficient distribution between soil spectral data and SOM content of three soil types.

### 3.2. Results of Characteristic Wavelength Screening

For the R spectra of the Paddy soil, the screening results based on the UVE algorithm is shown in Figure 6. A total of 815 wavelengths were screened from 2141 wavelengths, accounting for 37.89% of the total number of spectral wavelengths. The screened 551 wavelengths were distributed at 1223–1550 nm, 1929–2100 nm and 2233–2485 nm, whereas the 264 visible wavelengths were distributed at 355–432 nm, 506–610 nm and 637–718 nm.

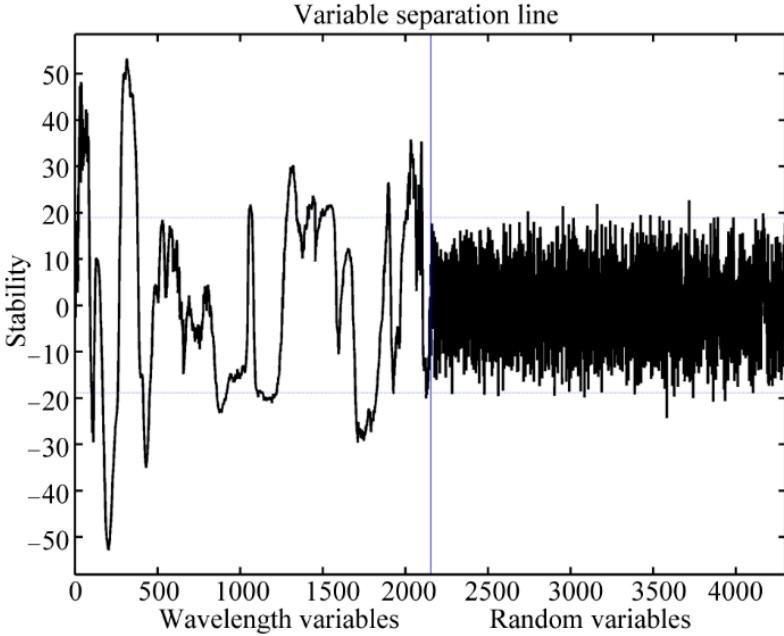

**Figure 6.** Key variables selected by UVE of the original spectral data of Paddy soil.

For the R spectra of Paddy soil, the screening results based on the CARS algorithm are shown in Figure 7. The number of screened wavelengths decreased continuously until reaching zero during the screening process, whereas the Monte Carlo sampling times or operation times increased continuously (Figure 7a). According to the trend graph of the RMSE of cross-validation (RMSECV) (Figure 7b), the modeling precision increased, whereas the RMSECV decreased when the operation time increased from 1 to 27 due to the deletion of the wavelengths which were poorly correlated with SOM. At the 27th sampling time,

RMSECV reached a minimum; therefore, the selected spectral variable subset was optimal. A total of 61 wavelengths screened by the CARS algorithm were mainly distributed within 1990–2490 nm, accounting for 2.84% of the total number of wavelengths.

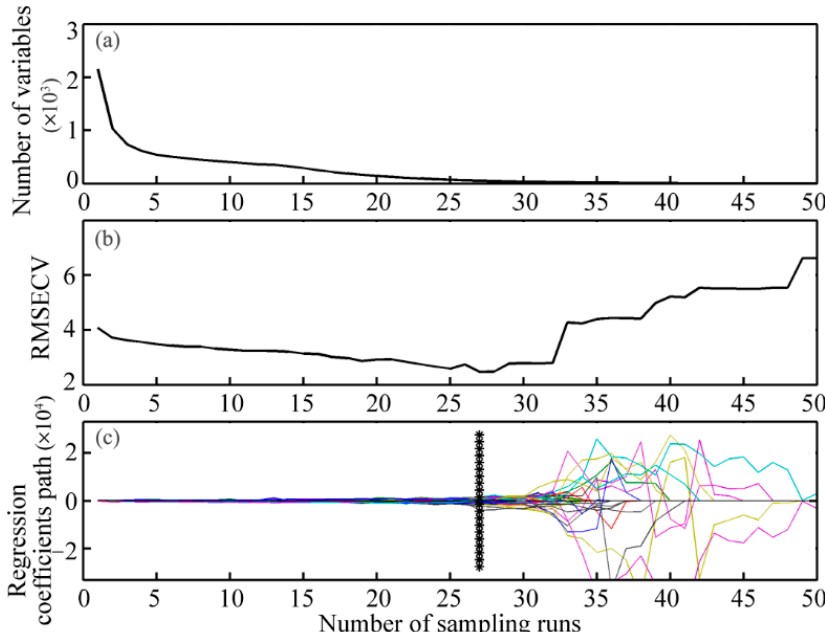

**Figure 7.** Key variables selected by CARS of Paddy soil original spectral data. (**a**) The number of screened wavelengths decreased continuously until reaching zero during the Monte Carlo sampling times increased. (**b**) RMSECV decreased when the Monte Carlo sampling times increased from 1 to 27. (**c**) Path of regression coefficients changes for variables in each sampling process. When at the 27th sampling times, RMSECV reached a minimum; therefore, the selected spectral variable subset was optimal.

The screened characteristic wavelengths of the two soil types are shown in Figure 8. The number of characteristic wavelengths using the UVE algorithm was higher than that with the CARS algorithm; this was related to the principles of the algorithms. The UVE algorithm screened 105–884 characteristic wavelengths for two soil types. The CARS algorithm compressed the characteristic wavelengths of the two soil types to lower than 6% of the full spectral wavelengths and reduced the complexity of the SOM spectral modeling. For the R, LR, CR and FDR spectra, the CARS algorithm screened 61–125 characteristic wavelengths from all 2141 wavelengths of Paddy soil and 40–61 for Shajiang black soil, respectively.

### 3.3. PLSR Modeling Based on Characteristic Wavelengths

The PLSR models of SOM were established using the selected wavelengths and full spectral wavelengths (Table 2). The validation results of the SOM PLSR models are shown in Figure 9 (Paddy soil) and Figure 10 (Shajiang black soil). For different soil types and spectral transformation data, the accuracy of the SOM models using the selected wavelengths was improved to different extents compared to the models using the full spectral wavelengths. The accuracy of the PLSR models combined with the CARS algorithm (CARS-PLSR) was higher than those of the PLSR models combined with the UVE algorithm (UVE-PLSR). CARS-PLRS models had the highest accuracies, with $R_p^2$, *RPD* and *LCCC* values higher than 0.80, 2.0 and 0.90, indicating that the SOM content could be accurately predicted. The PLSR modeling accuracy of paddy soil was better than that of Shajiang black soil.

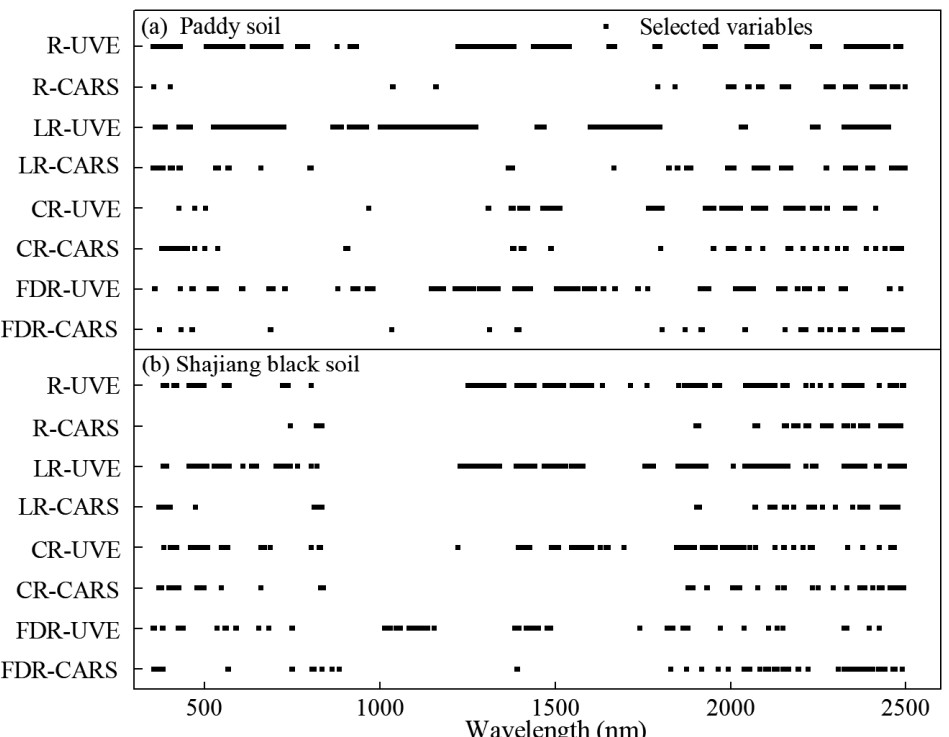

**Figure 8.** Plots of screened wavelengths based on UVE and CARS algorithms.

**Table 2.** PLSR models for SOM content based on selected wavelengths and spectral transformation datasets.

| Soil Type | Model [a] | Number of Wavelengths | Calibration Sets | | Validation Sets | | RPD | LCCC |
|---|---|---|---|---|---|---|---|---|
| | | | $R_c^2$ | $RMSE_c$ (g/kg) | $R_p^2$ | $RMSE_p$ (g/kg) | | |
| Paddy soil | R-F-PLSR | Full spectra | 0.83 | 2.86 | 0.76 | 3.66 | 2.06 | 0.88 |
| | R-UVE-PLSR | 815 | 0.92 | 1.99 | 0.81 | 3.29 | 2.29 | 0.91 |
| | R-CARS-PLSR | 61 | 0.91 | 2.09 | 0.87 | 2.68 | 2.81 | 0.93 |
| | LR-F-PLSR | Full spectra | 0.95 | 1.62 | 0.80 | 3.37 | 2.24 | 0.90 |
| | LR-UVE-PLSR | 884 | 0.90 | 2.16 | 0.86 | 2.77 | 2.72 | 0.92 |
| | LR-CARS-PLSR | 125 | 0.95 | 1.58 | 0.90 | 2.43 | 3.01 | 0.95 |
| | CR-F-PLSR | Full spectra | 0.70 | 3.81 | 0.62 | 4.66 | 1.62 | 0.79 |
| | CR-UVE-PLSR | 268 | 0.88 | 2.38 | 0.87 | 2.77 | 2.72 | 0.92 |
| | CR-CARS-PLSR | 70 | 0.97 | 1.21 | 0.95 | 1.72 | 4.31 | 0.97 |
| | FDR-F-PLSR | Full spectra | 0.92 | 1.96 | 0.78 | 3.51 | 2.15 | 0.87 |
| | FDR-UVE-PLSR | 300 | 0.88 | 2.38 | 0.83 | 3.09 | 2.44 | 0.91 |
| | FDR-CARS-PLSR | 70 | 0.91 | 2.03 | 0.94 | 1.81 | 4.18 | 0.97 |
| Shajiang black soil | R-F-PLSR | Full spectra | 0.85 | 1.53 | 0.58 | 2.44 | 1.55 | 0.72 |
| | R-UVE-PLSR | 366 | 0.82 | 1.67 | 0.69 | 2.10 | 1.80 | 0.80 |
| | R-CARS-PLSR | 40 | 0.94 | 0.98 | 0.79 | 1.72 | 2.19 | 0.87 |
| | LR-F-PLSR | Full spectra | 0.89 | 1.31 | 0.61 | 2.37 | 1.59 | 0.74 |
| | LR-UVE-PLSR | 461 | 0.84 | 1.57 | 0.62 | 2.34 | 1.61 | 0.76 |
| | LR-CARS-PLSR | 40 | 0.95 | 0.93 | 0.86 | 1.44 | 2.62 | 0.92 |
| | CR-F-PLSR | Full spectra | 0.92 | 1.10 | 0.28 | 3.21 | 1.17 | 0.54 |
| | CR-UVE-PLSR | 257 | 0.81 | 1.72 | 0.64 | 2.26 | 1.61 | 0.79 |
| | CR-CARS-PLSR | 53 | 0.92 | 1.11 | 0.73 | 1.96 | 1.93 | 0.84 |
| | FDR-F-PLSR | Full spectra | 0.96 | 0.78 | 0.26 | 3.23 | 1.45 | 0.55 |
| | FDR-UVE-PLSR | 105 | 0.82 | 1.67 | 0.63 | 2.30 | 1.94 | 0.79 |
| | FDR-CARS-PLSR | 61 | 0.98 | 0.55 | 0.85 | 1.46 | 2.58 | 0.92 |

Note: [a] R, LR, CR, and FDR stand for different spectral data. F stands for full spectral wavelengths; UVE stands for selected wavelengths by UVE algorithm; CARS stands for selected wavelengths by CARS algorithm. Model R-F-PLSR stands for PLSR model using all spectral wavelength reflectance; R-UVE-PLSR stands for PLSR modeling using selected reflectance wavelength by UVE algorithm.

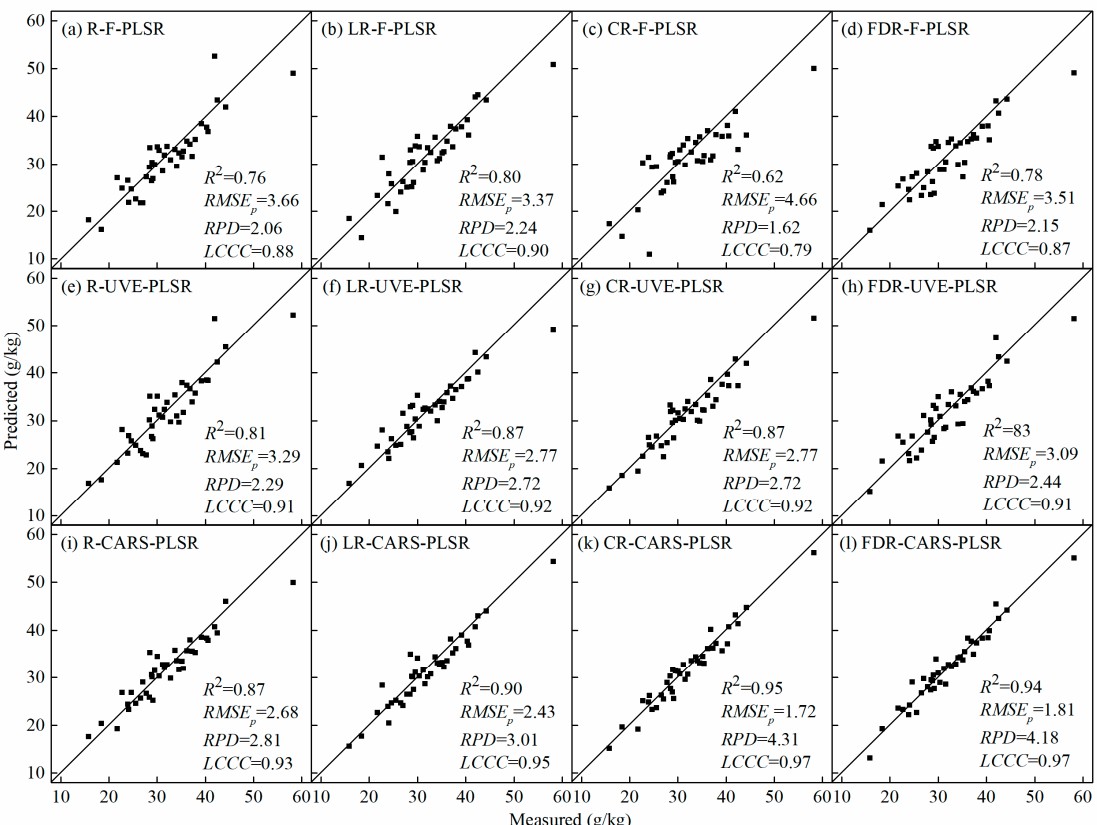

**Figure 9.** Scatter plots of measured and predicted SOM of Paddy soil.

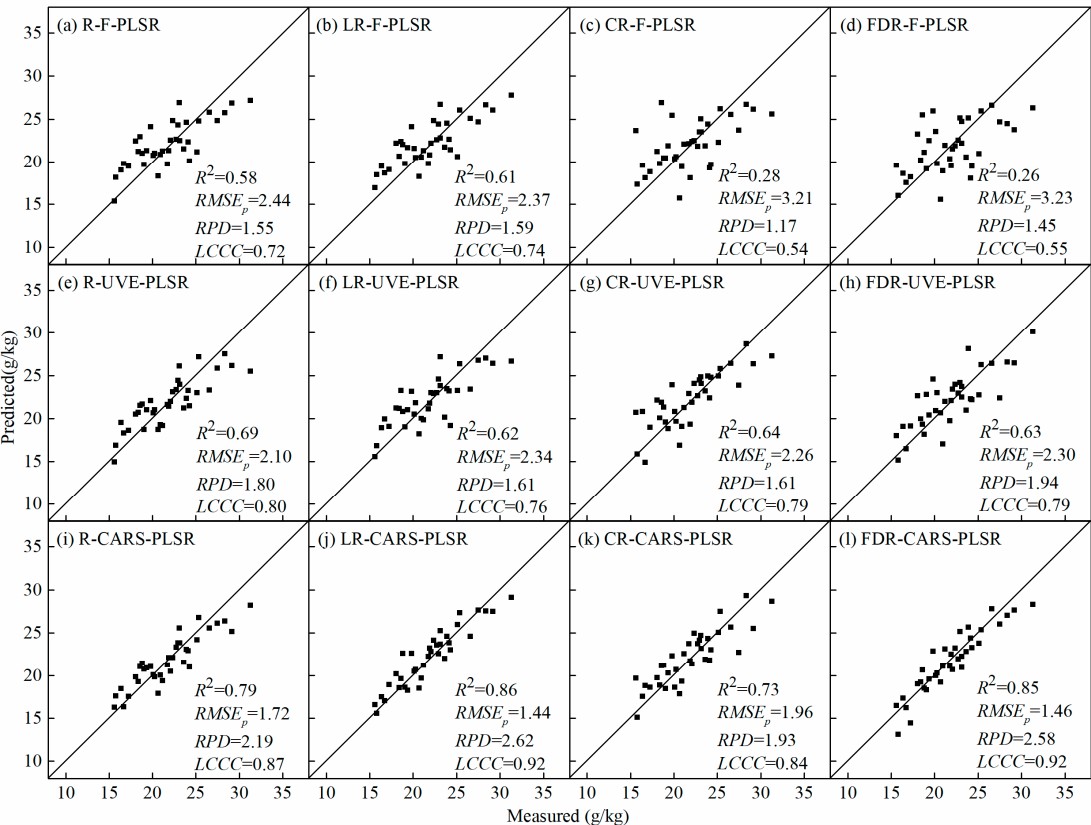

**Figure 10.** Scatter plots of measured and predicted SOM of Shajiang black soil.

With the same soil type and wavelength screening algorithm, the accuracy of the PLSR models after spectral transformation (LR, CR, and FDR) was improved compared with the original reflectance (R). For example, the CARS-PLSR models of Paddy soil (model LR-CARS-PLSR, CR-CARS-PLSR and FDR-CARS-PLSR), with $R_p^2$ and *RPD* values greater than 0.90 and 3 and $RMSE_p$ lower than 2.43 g/kg, outperformed model R-CARS-PLSR (with $R_p^2$ value of 0.87, *RDP* value of 2.81 and $RMSE_p$ value of 2.68 g/kg). *LCCC* values increased. For Shajiang black soil, the accuracy of LR-CARS-PLSR and FDR-CARS-PLSR was slightly better than that of R-CARS-PLSR. Overall, this study showed that LR and FDR transformation improved the modeling accuracy, which was consistent with other research results [23,35,36,38].

The improvement of model accuracy by characteristic wavelength screening was superior to that of spectral transformation. For example, in Paddy soil, the $R_p^2$ values of the PLSR models using full transformed spectra (model LR-F-PLSR, CR-F-PLSR and FDR-F-PLSR) ranged from 0.62 to 0.80, and the *RPD* values ranged from 1.62 to 2.24. Compared with R-F-PLSR ($R_p^2$ and *RPD* value of was 0.76 and 2.06), the accuracies of LR-F-PLSR, CR-F-PLSR and FDR-F-PLSR were slightly improved (Table 2). In Paddy soil, the $R_p^2$ and *RPD* value of the UVE-PLSR (LR-UVE-PLSR, CR-UVE-PLSR and FDR-UVE-PLSR) and CARS-PLSR models (LR-CARS-PLSR, CR-CARS-PLSR and FDR-CARS-PLSR) ranged from 0.81 to 0.95 and from 2.29 to 4.31, respectively, and the accuracy of these models were significantly improved than above models.

For two soil types, the predictive accuracy of the samples with SOM content lower than 20 g/kg was improved significantly in the CARS-PLSR models. The predicted and measured values were concentrated around the 1:1 line (Figures 9i–l and 10i–l). In this study area, the average SOM content of the Shajiang black soil was 21.60 ± 3.94 g/kg and the SOM content of 34% in the samples was lower than 20 g/kg (Table 1). For models using full spectral data of Shajiang black soil, the predicted and measured values were relatively dispersed near the 1:1 line, regardless of whether the SOM content was lower or higher than 20 g/kg (Figure 10a–d). The corresponding *LCCC* values were between 0.54 and 0.74 and the *RPD* values ranged from 1.17 to 1.59, indicating the poor predictive ability of the models. The accuracy of the CARS-PLSR models was significantly improved. The corresponding *LCCC* values were between 0.84 and 0.92 and the predicted and measured values were uniformly distributed near the 1:1 line (Figure 10i–l). The *RPD* values were all higher than 2, indicating the high accuracy and superior predictive capabilities of the models.

According to previous studies, soil spectral characteristics are not recommended for modeling and prediction when SOM is lower than 20 g/kg [48,49]. This might be because when there is a low SOM content, the spectral reflectance of the soils is dominated by other factors [48,50]. This study established optimal SOM prediction models for Shajiang black soil after CARS algorithm, suggesting that the CARS algorithm is an effective means to improve prediction precision for soils with low SOM contents.

### 3.4. SVR and RF Modeling Based on Characteristic Wavelengths

The SVR and RF models of SOM were established based on the characteristic wavelengths and the full spectra data (Tables 3 and 4). For different soil types and spectral transformation data, the results of SVR modeling were superior to those of RF modeling. On the whole, the accuracy of the SVR models was similar to that of PLSR models, while the accuracy of the RF models was notably worse than that of PLSR models. RF models performed worse for two soil types, with the $R^2$ values of validation ranging from 0.22 to 0.68 and *RPD* values ranging from 1.01 to 1.60. The SVR and RF modeling accuracies of Paddy soil were better than that of Shajiang black soil.

**Table 3.** SOM modeling results based on SVR using selected wavelengths and spectral transformation datasets.

| Soil Type | Model [a] | Best Parameters | Calibration Sets | | Validation Sets | | RPD | LCCC |
|---|---|---|---|---|---|---|---|---|
| | | (C) | $R_c^2$ | $RMSE_c$ (g/kg) | $R_p^2$ | $RMSE_p$ (g/kg) | | |
| Paddy soil | R-F-SVR | 16 | 0.99 | 0.66 | 0.75 | 4.03 | 1.87 | 0.85 |
| | R-UVE-SVR | 2 | 0.81 | 3.09 | 0.77 | 3.93 | 1.92 | 0.85 |
| | R-CARS-SVR | 16 | 0.89 | 2.57 | 0.88 | 2.85 | 2.70 | 0.92 |
| | LR-F-SVR | 4 | 0.95 | 1.60 | 0.83 | 3.20 | 2.36 | 0.90 |
| | LR-UVE-SVR | 16 | 0.93 | 1.91 | 0.89 | 2.57 | 2.93 | 0.93 |
| | LR-CARS-SVR | 16 | 0.94 | 1.72 | 0.92 | 2.33 | 3.24 | 0.95 |
| | CR-F-SVR | 0.0625 | 0.99 | 0.68 | 0.79 | 3.56 | 2.12 | 0.86 |
| | CR-UVE-SVR | 0.0625 | 0.90 | 2.20 | 0.85 | 3.08 | 2.45 | 0.91 |
| | CR-CARS-SVR | 8 | 0.98 | 1.04 | 0.91 | 2.24 | 3.37 | 0.96 |
| | FDR-F-SVR | 0.0625 | 0.99 | 0.65 | 0.76 | 3.96 | 1.90 | 0.86 |
| | FDR-UVE-SVR | 0.0625 | 0.95 | 1.56 | 0.78 | 3.61 | 2.09 | 0.88 |
| | FDR-CARS-SVR | 0.0625 | 0.93 | 1.94 | 0.91 | 2.37 | 3.18 | 0.94 |
| Shajiang black soil | R-F-SVR | 1 | 0.91 | 1.23 | 0.63 | 2.30 | 1.64 | 0.76 |
| | R-UVE-SVR | 8 | 0.87 | 1.46 | 0.66 | 2.20 | 1.71 | 0.79 |
| | R-CARS-SVR | 16 | 0.94 | 1.00 | 0.77 | 1.79 | 2.10 | 0.86 |
| | LR-F-SVR | 1 | 0.92 | 1.17 | 0.61 | 2.36 | 1.60 | 0.75 |
| | LR-UVE-SVR | 2 | 0.79 | 1.81 | 0.70 | 2.09 | 1.80 | 0.81 |
| | LR-CARS-SVR | 16 | 0.94 | 1.00 | 0.82 | 1.58 | 2.38 | 0.90 |
| | CR-F-SVR | 0.0625 | 0.99 | 0.37 | 0.29 | 3.20 | 1.18 | 0.48 |
| | CR-UVE-SVR | 0.0625 | 0.95 | 0.91 | 0.63 | 2.34 | 1.61 | 0.77 |
| | CR-CARS-SVR | 1 | 0.97 | 0.71 | 0.69 | 2.16 | 1.74 | 0.83 |
| | FDR-F-SVR | 0.0625 | 0.99 | 0.38 | 0.36 | 3.07 | 1.23 | 0.58 |
| | FDR-UVE-SVR | 0.0625 | 0.91 | 1.18 | 0.67 | 2.23 | 1.69 | 0.77 |
| | FDR-CARS-SVR | 0.0625 | 0.99 | 0.45 | 0.83 | 1.58 | 2.38 | 0.91 |

Note: [a] R, LR, CR, and FDR stand for different spectral data. F stands for full spectral wavelengths; UVE stands for selected wavelengths by UVE algorithm; CARS stands for selected wavelengths by CARS algorithm. Model R-F-SVR stands for SVR model using full spectral wavelength reflectance data; R-UVE-SVR stands for SVR modeling using selected reflectance wavelength by UVE algorithm.

The accuracies of the SVR models combined with the CARS algorithm (CARS-SVR) and UVE algorithm (UVE-SVR) were higher than those of the SVR models using full spectral data, and CARS-SVR models were best. In Paddy soil, the $R_p^2$ values of the CARS-SVR models ranged from 0.88 to 0.92, the *RPD* values ranged from 2.70 to 3.37 and the *LCCC* value ranged from 0.92 to 0.96 (Table 3). The $R_p^2$ values of SVR models using full spectral data ranged from 0.75 to 0.83, the *RPD* values ranged from 1.87 to 2.36 and the *LCCC* value ranged from 0.85 to 0.90. These results indicated that a combination of the UVE and CARS algorithms could significantly improve the accuracy of SVR modeling.

For the R spectra of Paddy soil, LR, CR and FDR improved the SVR and RF modeling accuracy moderately. For example, the CARS-SVR models of Paddy soil (model LR-CARS-SVR, CR-CARS-SVR and FDR-CARS-SVR), with $R_p^2$ and *RPD* values greater than 0.90 and 3 and $RMSE_p$ lower than 2.37 g/kg, outperformed R-CARS-PLSR (with $R_p^2$ value of 0.88, *RDP* value of 2.70, $RMSE_p$ value of 2.85 g/kg). *LCCC* values increased moderately. Additionally, the UVE-SVR models of Paddy soil (model LR-UVE-SVR, CR-UVE-SVR and FDR-UVE-SVR) also outperformed R-UVE-SVR moderately. In Shajiang black soil, spectral transformation could not improve the SVR and RF modeling accuracy.

The results showed that the improvement of SVR and PLSR modeling accuracy by characteristic wavelength screening was superior to that of spectral transformation. CARS-PLSR and CARS-SVR using CR spectra produced the best predictions (hightest $R^2$ and *RPD*, lowest *RMSE*) for SOM modeling of Paddy soil. CARS-PLSR and CARS-SVR using LR and FDR spectra were optimal for Shajiang black soil.

**Table 4.** SOM modeling results based on RF model using selected wavelengths and spectral transformation datasets.

| Soil Type | Model [a] | Best Parameters | Calibration Sets | | Validation Sets | | RPD | LCCC |
|---|---|---|---|---|---|---|---|---|
| | | (*mtry, ntree*) | $R_c^2$ | $RMSE_c$ (g/kg) | $R_p^2$ | $RMSE_p$ (g/kg) | | |
| Paddy soil | R-F-RF | 9, 200 | 0.43 | 5.28 | 0.51 | 5.44 | 1.39 | 0.63 |
| | R-UVE-RF | 10, 500 | 0.52 | 5.19 | 0.35 | 5.30 | 1.10 | 0.63 |
| | R-CARS-RF | 6, 100 | 0.43 | 5.65 | 0.25 | 5.78 | 1.17 | 0.45 |
| | LR-F-RF | 8, 100 | 0.40 | 5.45 | 0.48 | 5.60 | 1.35 | 0.60 |
| | LR-UVE-RF | 10, 1500 | 0.43 | 5.26 | 0.48 | 5.50 | 1.37 | 0.63 |
| | LR-CARS-RF | 10, 100 | 0.47 | 5.07 | 0.50 | 5.56 | 1.36 | 0.58 |
| | CR-F-RF | 6, 200 | 0.47 | 5.16 | 0.64 | 5.05 | 1.49 | 0.65 |
| | CR-UVE-RF | 10, 100 | 0.59 | 4.88 | 0.34 | 5.48 | 1.02 | 0.69 |
| | CR-CARS-RF | 8, 100 | 0.52 | 5.22 | 0.50 | 4.69 | 1.01 | 0.70 |
| | FDR-F-RF | 7, 100 | 0.52 | 4.86 | 0.65 | 4.80 | 1.57 | 0.70 |
| | FDR-UVE-RF | 10, 100 | 0.53 | 4.80 | 0.68 | 4.70 | 1.60 | 0.71 |
| | FDR-CARS-RF | 6, 100 | 0.51 | 4.87 | 0.66 | 4.76 | 1.57 | 0.70 |
| Shajiang black soil | R-F-RF | 1, 1000 | 0.07 | 4.05 | 0.30 | 3.14 | 1.20 | 0.46 |
| | R-UVE-RF | 5, 100 | 0.08 | 4.06 | 0.26 | 3.26 | 1.19 | 0.48 |
| | R-CARS-RF | 2, 100 | 0.10 | 3.96 | 0.22 | 0.52 | 1.12 | 0.36 |
| | LR-F-RF | 1, 500 | 0.06 | 4.08 | 0.28 | 3.19 | 1.18 | 0.45 |
| | LR-UVE-RF | 7, 200 | 0.04 | 4.17 | 0.31 | 3.13 | 1.20 | 0.48 |
| | LR-CARS-RF | 1, 200 | 0.08 | 3.95 | 0.20 | 3.37 | 1.12 | 0.34 |
| | CR-F-RF | 10, 100 | 0.12 | 3.75 | 0.14 | 3.51 | 1.07 | 0.20 |
| | CR-UVE-RF | 8, 100 | 0.21 | 3.56 | 0.28 | 3.28 | 1.15 | 0.33 |
| | CR-CARS-RF | 8, 200 | 0.20 | 3.64 | 0.21 | 3.41 | 1.09 | 0.26 |
| | FDR-F-RF | 10, 100 | 0.30 | 3.39 | 0.51 | 2.84 | 1.33 | 0.53 |
| | FDR-UVE-RF | 4, 100 | 0.52 | 2.84 | 0.46 | 2.79 | 1.40 | 0.61 |
| | FDR-CARS-RF | 10, 100 | 0.56 | 2.93 | 0.60 | 2.73 | 1.38 | 0.56 |

Note: [a] R, LR, CR, and FDR stand for different spectral data. F stands for all spectral wavelengths; UVE stands for selected wavelengths by UVE algorithm; CARS stands for selected wavelengths by CARS algorithm. Model R-F-RF stands for RF model using all spectral wavelength reflectance data; R-UVE-RF stands for RF modeling using selected reflectance wavelength by UVE algorithm.

## 4. Discussion

This study selected effective spectral wavelengths using the CARS and UVE algorithms. The two algorithms decreased the number of input variables for modeling and increased the modeling accuracy and robustness. In this study, the CARS algorithm reduced the number of wavelengths from the original 2141 to 40–125 for R, LR, CR and FDR, and the UVE algorithm selected 257–884 wavelengths from the full spectral data of two soil types. The screened wavelengths were mainly distributed in the ranges of 400–900 nm, 1400–1700 nm and 2000–2400 nm, which was consistent with the research conclusions of Yu et al. (2016) [32], Tang et al. (2021) [33] and Bao et al. (2020) [26]. This further proved the importance of eliminating uninformative variables from full spectral data during spectral modeling [26–28,32,33].

After combining with the CARS algorithm, the modeling precision were remarkably improved compared to models combined with UVE algorithm. This was consistent with the studies reported by Vohland et al., Yu et al. and Tang et al. [27,32,33]. The CARS algorithm was superior to the UVE algorithm during SOM spectral modeling, which was mainly related to the different principles of the two algorithms. The CARS algorithm selects variables with relatively high absolute regression coefficient values in the PLSR model based on adaptive reweighted sampling technology and eliminates wavelengths with small weights [29]. The UVE algorithm selects variables based on the stability of the PLSR coefficient. It can avoid model overfitting and increase the prediction abilities of the models [43,44]. This approach differs from previous wavelength selection methods

(i.e., according to prior knowledge or the correlation with SOM). For example, in the R spectral data of Paddy soil, a total of 61 wavelengths selected by the CARS algorithm were mainly distributed within 1990–2495 nm, with the absolute value of correlation coefficient between reflectance and SOM content being lower than 0.46.

SVR and RF methods have advantages over other prediction models, as they are able to model complex, non-linear and linear relationships between variables [10]. Rossel and Behrens [10] reported that predictions by SVR using all VNIR wavelengths produced the smallest RMSE values, with RF performing weakly. Ji et al. [25] reported that SVR using all VNIR wavelengths produced the best prediction ($R^2$ of validation: 0.64 and *RPD*: 2.16.), while the precision of RF was poor ($R^2$ of validation: 0.40 and *RPD*: 1.61). Terra et al. (2015) [51] also modeled SOM accurately based on SVR with a linear kernel function. Dotto et al. [23] found that the SVR model yielded robust predictions while the overall predictive ability of RF models was considered insufficient. The $R^2$ of RF model was 0.47 to 0.77. Our study was consistent with the above studies, and proved that the UVE and CARS algorithms can improve SVR modeling accuracy, even though the accuracy is poor when combined with the RF model. For example, Paddy soil SVR modeling produced greater $R^2$ and *RPD* values, from 0.75 to 0.92 and from 1.87 to 3.24.

However, Knox et al. [52] showed that the RF model produced an $R^2$ from 0.63 to 0.88 when using different spectral preprocessing only in the VNIR range. The RF model combined with CARS produced more accurate SOM predictions, with $R^2$ values ranging from 0.65 to 0.89, as reported by Bao et al. [26]. In that study, the SOM content ranged from 4.25 to 80.32 g/kg, with a mean of 39.5 ± 13.21 g/kg. In our study, the RF models performed worse for two soil types, with the $R^2$ values ranging from 0.22 to 0.68, *RPD* values ranging from 1.01 to 1.60 and mean SOM contents of 32.13 ± 7.21 g/kg and 21.60 ± 3.94 g/kg for Paddy soil and Shajiang black soil, respectively. The difference in soil types and SOM content levels might be a reason that RF models performed differently.

The PLSR and SVR models for Shajiang black soil using full spectral domain of R, LR, CR and FDR produced poor results. The $R^2$ in validation ranged from 0.26 to 0.63 and the *RPD* ranged from 1.17 to 1.64. The poor model performance was consistent with the conclusions of Lu et al. [53]; their research was mainly related to low SOM content and its weak correlation with the spectra. In this study, there were no significant differences in the correlations between the SOM and different wavelengths, with the absolute value of the correlation coefficient ranging between 0.30 and 0.48 (Figure 5b). Moreover, there were few differences in the characteristics among the spectral curves of different SOM contents (Figure 4b). After screening the wavelengths with the CARS algorithm, the PLSR and SVR models were significantly improved, with *RPD* values greater than 2.0.

For soils with low SOM contents, different spectral transformation approaches can improve the precision of spectral modeling. Nawar et al. [35] reported that CR and FDR spectral transformation improved SOM prediction models, to varying degrees, based on PLSR, SVR and MARS. In that study, the SOM content ranged between 0.2 and 23.0 g/kg, averaging 8.9 g/kg. Wang et al. [54] found that discrete wavelet transformation of the original spectra improved the modeling precision of SOM in northern China. The $R^2$ of the optimal model reached as high as 0.72. In that study, the average SOM was 15.76 g/kg and about 70% of samples had SOM contents lower than 20 g/kg. Yang et al. [55] used spectral characteristic indexes to efficiently predict SOM content of Shajiang black soil in the eastern China, achieving a high degree of precision ($R^2$: 0.81). That study included 45 soil samples with SOM contents ranging from 2.07 g/kg to 21.21 g/kg. Further spectral processing, wavelength screening algorithms and modeling techniques were applied to hyperspectral modeling of soil properties. For different soil types and SOM content levels, the optimal spectral treatment might be different, although this requires more comparative studies in the future.

## 5. Conclusions

(1) The CARS and UVE algorithms reduced the extent of the soil hyperspectral data and the complexity of SOM spectral modeling. The CARS algorithm had a relatively high compression ratio and selected 40–125 characteristic wavelengths from all VNIR wavelengths of R, LR, CR and FDR. The selected wavelengths of the two soil types were mainly distributed in the near-infrared wavelength range.

(2) For the two soil types and four full spectral domains (R, LR, CR, and FDR), the CARS and UVE algorithms improved the SOM modeling precision based on the PLSR and SVR methods. PLSR and SVR combined with the CARS algorithm displayed the best prediction power, providing an important reference for band selection. The improvement of SVR and PLSR modeling accuracy by CARS and UVE was superior to that of spectral transformation.

(3) CARS-PLSR and CARS-SVR using CR spectra produced the best predictions (highest $R^2$ and *RPD*, lowest *RMSE*) for SOM modeling of Paddy soil. CARS-PLSR and CARS-SVR using LR and FDR spectra were the optimal models for Shajiang black soil. The modeling accuracies of PLSR and SVR of Paddy soil were better than those for Shajiang black soil. RF models performed poorly for both soil types. The CARS algorithm improved predictions considerably for soil samples with low SOM contents.

**Author Contributions:** Conceptualization, M.Z.; Investigation, Y.L.; Software, Y.G.; Validation, Y.G.; Visualization, Y.L.; Writing—original draft, M.Z.; Writing—review & editing, M.Z. and S.W. All authors have read and agreed to the published version of the manuscript.

**Funding:** This research was funded by the National Natural Science Foundation of China, grant numbers 41501226 and 31700369; Natural Science Foundation of the Higher Education Institutions of Anhui Province, grant number KJ2015A034; Research Fund for Doctoral Program of Anhui University of Science and Technology, grant number ZY020, respectively.

**Conflicts of Interest:** The authors declare no conflict of interest.

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
