# Peer review of "Hyperspectral Modeling of Soil Organic Matter Based on Characteristic Wavelength in East China"

_sustainability, doi:10.3390/su14148455_

Round 1

Reviewer 1 Report

The study seeks to relate Soil Organic Matter (SOM) to reflectance via spectroscopy. While the study is of interest, there are a number of areas, listed below, which require improvement either in description or in the experimental design. 

1) From the title, one expects that hyperspectral imaging was conducted. However, from the description in the manuscript, it becomes clear that the methodology used was spectroscopy. Without a more detailed description of the experiments and the rationale behind them, it is difficult to judge whether the experimental design e.g. the equipment used (Lines 144-145) and the experimental environment (Line 147), were appropriate for the study's objectives. 

2) Further, with regards to the description of the experiments, the information provided in Lines 137-142 is insufficient in terms of some of the inferences or attributions made throughout the manuscript. For example, in Lines 233-236, reference is made to soil properties for which there is no information on testing for them i.e. moisture content, and kaolinites. This leads to inferences and conclusions about the soil spectral response that may not be valid e.g. Lines 233-234.

3) Lines 215-223: Kindly elaborate on why the 15 - 20 g/Kg SOM level category has not been captured in reporting and discussing the results. This range appears to have the highest reflectance in Figure 2(a). Therefore the attribution made in Lines 220-223 may not hold since this category had lower SOM than the 20-25 g/Kg SOM level class, yet it showed the highest reflectance. 

3) Kindly provide results of the actual laboratory-measured SOM content reported prior to relating SOM content to spectral response. 

Reviewer 2 Report

The manuscript titled “ Hyperspectral Modeling of Soil Organic Matter Based on Characteristic Wavelength and Spectral Transformation in East China“. I find the work interesting and in line with the aim of the journal. I have some concerns about the experimental set-up to justify what the authors claim. Moreover, the rationale behind some of the data presented was not entirely clear. I also recommend to the authors to improve their references by conducting a more extensive review on international literature.  

Ø  I suggest to modify the title title should be more crisp and brief

Ø  Abstract introductory statement is too long, it has to be improved with a more specific rationale of the study. The abstract should have crisp information about the aim meterials method result and conclusion, which I don't find. 

Ø  Result and Discussion is more like explaining results, at most of the places I missed connection with past studies, and proper explanation is also missing. Very few citations are in the discussion section.

Ø  Above are a few suggestions, especially the author should modify the abstract it looks like long story rest is ok. I congratulate the author for a great pice of work.

Reviewer 3 Report

(1)This study evaluated the performance of different feature selection methods, four spectral forms, and three calibration methods in SOM estimation.

(2)Introduction: Line 109-110: “However, there are few comparisons of the improvement of model accuracy by spectral transformation and characteristic wavelength screening.”

This description is partial. I did some literature searching. Please give a more objective summary of the spectral transformation and the calibration methods.

e.g. “Xie S, Ding F, Chen S, Wang X, Li Y, Ma K. Prediction of soil organic matter content based on characteristic band selection method. Spectrochim Acta A Mol Biomol Spectrosc. 2022 May 15;273:120949. doi: 10.1016/j.saa.2022.120949. Epub 2022 Jan 26. PMID: 35183857.”

The authors listed some comparisons of different spectral transformations.

“Shi Y, Zhao J, Song X, et al. Hyperspectral band selection and modeling of soil organic matter content in a forest using the Ranger algorithm. PLoS One. 2021;16(6):e0253385. Published 2021 Jun 28. doi:10.1371/journal.pone.0253385”

The authors listed a summary of spectral transformations and calibration methods.

(3)Section Results: In this section, some discussions of the spectral features and the comparison of the soil spectra in this study and other research were included. eg. line 220-223, 230-236,244-251. 296-303.

I suggest put these sentences in the discussion section and should be shortened.

Line 213-214. Please give the reference for this interval division.

Round 2

Reviewer 1 Report

Some improvements have been made in terms of the description of the experimental design and methods thus significantly improving the quality of the manuscript. 

Please ensure that the text and figures are consistent e.g. Line 236 references Figure 2(b) yet this is not the case.

Author Response

Thank you for your comments.

"Line 236 references Figure 2(b) yet this is not the case." It is Figure 3b. I revised in the new version.